Somatic mutation profiling, tumor-infiltrating leukocytes, tertiary lymphoid structures and PD-L1 protein expression in HER2-amplified colorectal cancer

Liu Xiao-Ting 1
Kou Zhi-Yong 1
Zhang Hushan 2
Dong Jian 3
Zhang Jian-Hua 4
Peng You-Jun 4
Ma Shu Min 5
Liang Lei 4
Meng Xuan-Yu 1
Zhou Yuan 1
Yang Jun yangjun6@kmmu.edu.cn 1
1 Department of Oncology, The First Affiliated Hospital of Kunming Medical University , Kunming , Yunnan , China
2 Zhaotong Healthy Vocational College , Zhaotong , Yunnan , China
3 Colorectal Cancer Clinical Research Center, Yunnan Cancer Hospital , Kunming , Yunnan , China
4 Department of General Surgery, The Third People’s Hospital of Honghe Prefecture , Honghe , Yunnan , China
5 Department of General Surgery, The Second People’s Hospital of Qujing , Qujing , Yunnan , China
Coates Philip
Electronic publication date: 2023 May 2
Publication date: 2023
Volume: 11
Electronic Location ID: e15261
Received 2023 Jan 12; Accepted 2023 Mar 28
Copyright: ©2023 Liu et al.
Copyright year: 2023
Copyright holder: Liu et al.
License: This is an open access article distributed under the terms of the Creative Commons Attribution License, which permits unrestricted use, distribution, reproduction and adaptation in any medium and for any purpose provided that it is properly attributed. For attribution, the original author(s), title, publication source (PeerJ) and either DOI or URL of the article must be cited.
License URL: https://creativecommons.org/licenses/by/4.0/

Keywords: Colorectal cancer, HER2, Tumour immune microenvironment, PD-L1 expression, Somatic mutation profiling

Funding: National Natural Science Foundation of China 82160533 Applied Basic Foundation of Yunnan Province 202001AT070009 Yunnan Health Training Project of High Level Talents D-2019032 535 Talent Project of First Affiliated Hospital of Kunming Medical University 2022535D07 Yunnan Revitalization Talent Support Program RLQB20200004 Graduate Innovation Fund of Kunming Medical University 2022S060 This work was supported by the National Natural Science Foundation of China (No. 82160533), the Applied Basic Foundation of Yunnan Province (No. 202001AT070009), the Yunnan Health Training Project of High Level Talents (No. D-2019032), the 535 Talent Project of First Affiliated Hospital of Kunming Medical University (No. 2022535D07), the Yunnan Revitalization Talent Support Program (No. RLQB20200004) and the Graduate Innovation Fund of Kunming Medical University (No. 2022S060). The funders had no role in study design, data collection and analysis, decision to publish, or preparation of the manuscript.

==============================
The status of human epidermal growth factor receptor 2 (HER2) for the prognosis in colorectal cancer (CRC) is controversial, and the characteristics of the somatic mutation spectrum, tumor-infiltrating leukocytes, tertiary lymphoid structures and PD-L1 protein are unknown in HER2-amplified colorectal cancer (HACC). In order to explore these characteristics along with their correlation with clinicopathological factors and prognosis in HACC. Samples of 812 CRC patients was collected. After immunohistochemistry (IHC), 59 of 812 were found to be HER2-positive, then 26 of 59 samples were further determined to be HER2 amplification by fluorescence in situ hybridization (FISH). Somatic mutation profiling of HACC was analysed using whole exome sequencing (WES). Multiplex fluorescence immunohistochemistry (mIHC) was used for tumor-infiltrating leukocytes and tertiary lymphoid structures (TLSs), while PD-L1 protein was detected by IHC. Our results indicate that the detection rates of HER2 positivity by IHC and FISH were 7.3% and 3.2% respectively, and HER2 amplification is correlated with distant tumour metastasis. The somatic mutation profiling revealed no differences between HACC and HER2-negative CRC. However, TP 53 strongly correlated with poor prognosis in HACC. Furthermore, tumor-infiltrating T cells and TLSs in the tumor immune microenvironment, as well as PD-L1 expression, were higher in HACC than in HER2-negative controls. However, none of them were associated with the prognosis of HACC. In all, HER2 amplification is correlated with distant metastasis and TP53 gene mutation may be a potential protective mechanism of HACC.

Introduction

The activation of the human epidermal growth factor receptor-2 (HER2) gene, whose downstream signal transduction pathways include MAPK and PI3K/AKT/mTOR, leads to cell proliferation and differentiation, thus inhibiting apoptosis and tumor progression (Guarini et al., 2021).

HER2 is overexpressed in 7%–34% of gastric cancer, 13%–20% of breast cancer, 1.9%–14.3% of lung cancer, and 2%–11% of CRC cases (Guarini et al., 2021; Ivanova et al., 2022; Siena et al., 2018). However, the association between HER2-positive status and the prognosis of CRC is controversial (Huan et al., 2017; Huang et al., 2022; Shabbir et al., 2016; Ting & Xichen, 2022). For HER2-positive CRC patients, double anti-HER2 therapy is recommended (Kavuri et al., 2015; Oh & Bang, 2020; Vranic, Beslija & Gatalica, 2021), nevertheless, some patients have disease refractory to treatment. Unfortunately, refractory HER2-positive CRC is not widely studied, and the molecular mutation spectrum is poorly understood. Moreover, there is a lack of research on whether HER2-positive CRC has characteristic mutations that can be used as potential therapeutic targets (Imyanitov & Kuligina, 2021).

Immune checkpoint inhibitors (ICIs) are widely used in microsatellite instability-high (MSI-H) CRC, and their use in microsatellite-stable (MSS) CRC is being investigated in several clinical trials (Cui, 2022; Wang et al., 2021). However, there are no reports of ICIs being used clinically for the treatment of HER2-positive CRC.

Notably, PD-1/PD-L1, a popular immune checkpoint, is upregulated in various tumors and induces tumor immune escape (He et al., 2021; Yang et al., 2022). Nonetheless, whether the therapeutic effects of ICI treatment in HER2-positive CRC correlate with the programmed death-ligand 1 (PD-L1) protein expression and the status of the tumor microenvironment (TME) immune cells, including T cells, macrophages, natural killer (NK) cells, B cells, and tertiary lymphoid structures (TLSs), are unknown. TLSs are ectopic lymphoid organs formed by non-lymphoid tissues and are associated with a better prognosis in patients receiving immunotherapy (Vanhersecke et al., 2021).

This study aims to explore the relationship between HER2 amplification in CRC cases with varying clinicopathological factors and prognosis and gene mutations, tumor-infiltrating leukocytes, TLSs, and PD-L1 protein expression.

Patients and Methods

Patients

General basic information and surgical resection specimens of 812 patients with CRC were collected. Patients underwent surgical treatment from January 2014 to December 2021 in the First Affiliated Hospital of Kunming Medical University, Kunming, China, and their HER2 status was determined based on postoperative IHC.Collected patient information included age, sex, tumor location, tumor stage, lymph node, metastasis, Tumor Node Metastasis(TNM) stage, mismatch repair (MMR) protein status, the tumor marker carcinoembryonic antigen (CEA), and patient outcome data: overall survival (OS) and progression free survival (PFS). Inclusion criteria: 812 CRC cases based on clinicopathological information. Exclusion criteria: patients with incomplete basic information and follow-up information related to clinical pathology, patients lost follow-up, and patients with inadequate wax blocks in tissue specimens.

Ethics statement

This study is a single-centre clinical retrospective study, approved by and performed in accordance with advice of the Research Ethics Board of Kunming Medical University (Kunming, China) and exempt from signing the patient’s informed consent form (Ethical Application Ref: 2022L-271).

Patient characteristics

A total of 812 patients with CRC were included in this study, and 59 cases were confirmed by IHC as HER2-positive samples, among them, 18 cases were confirmed to be IHC- HER2 1+, 18 cases were confirmed to be IHC- HER2 2+, and 23 cases were confirmed to be IHC- HER2 3+. The above 59 samples were tested using fluorescence in situ hybridization (FISH), and 26 cases were proved to be HER2 amplification. Of the 26 cases, 15 cases were male and 11 were female; median age was 63 years (age range 26–79 years); pT stages: pT0: zero cases, pT1: zero cases, pT2: one case, pT3: 21 cases, pT4: four cases; pN0: 12 cases, pN1: 11 cases, pN2: three cases; pM0: 17 cases; pM1: nine cases. According to the eighth edition of the Union for International Cancer Control (UICC) staging standard of CRC, there was one case in stage I, nine cases in stage II, seven cases in stage III, and nine cases in stage IV. As for mismatch repair, there were three cases of deficient mismatch repair (dMMR) and 23 cases of proficient mismatch repair (pMMR). Using the colonic splenic curvature as the boundary, there were 21 cases of left CRC cases and five right colon cancer cases. Follow-up was until the first of May, 2022, or patient’s passing. The median survival duration was 25.5 months, with the longest follow-up being 69 months, and the shortest being 5 months. During follow-up, six patients had disease progression, four were stage IV patients, of which there was one relapse, one peritoneal metastasis, one lung and bone metastases, and one bone metastasis; there was one case each of peritoneal metastasis, and peritoneal and liver metastases in stage III and stage II patients, respectively. The propensity score matching (PSM) was used to select 26 patients from the remaining 786 CRC cases with no HER2 amplification to be used in the control group (Table 1).

Table 1 Clinicopathological characteristics of patients with HACC and HER2-negative colorectal cancer.

Factors	Total (n = 812)			
	HER2 amplification (n = 26)	HER2 negative (n = 786)	X2	P	
Age					
<63years	10 (38.46%)	432(54.96%)	2.763	0.096	
≥63years	16 (61.54%)	354(45.04%)	
Gender					
Male	15 (57.69%)	455 (57.89%)	0	0.984	
Female	11 (42.31%)	331 (42.11%)	
MMR					
pMMR	23 (88.46%)	610 (77.61%)	1.515	0.283	
dMMR	3 (11.54%)	176 (22.39%)	
CEA					
≤5	13 (50%)	406 (51.65%)	0.028	0.868	
>5	13 (50%)	380 (48.35%)	
TNM stages					
I	1 (3.85%)	46 (5.86%)	7.249	0.064	
II	9 (34.62%)	299 (38.04%)	
III	7 (26.92%)	320 (40.71%)	
IV	9 (34.61%)	121 (15.39%)	
T staging					
2	1 (3.85%)	94 (11.96%)	5.093	0.078	
3	21 (80.77%)	463 (58.91%)	
4	4 (15.38%)	229 (29.13%)	
N staging					
0	12 (46.15%)	384 (48.85%)	1.527	0.466	
1	11 (42.31%)	254 (32.32%)	
2	3 (11.54%)	148 (18.83%)	
M					
0	17 (65.38%)	665 (84.61%)	6.915	0.009	
1	9 (34.62%)	121 (15.39%)	
The primary site of the tumor					
Left	21 (80.77%)	611 (77.74%)	0.134	0.714	
Right	5 (19.23%)	175 (22.26%)	

Experimental procedure

Survival analysis was performed on 59 IHC-confirmed HER2-positive (IHC 1+, IHC 2+, IHC 3+) cases. From these cases, 26 samples were identified by FISH to have HER2 amplification and were used as the experimental group. The control group was obtained as described above using PSM, and the patients in both groups were followed up until May 2022 or the patients’ passing. Molecular mutations characteristic of HER2 amplification in CRC were analysed using whole exome sequencing (WES). Multiplex IHC (mIHC) was used to characterize, localize and quantify TME immune cells including T cells, macrophages, NK cells, B cells; as well as TLSs. PD-L1 expression was detected using IHC. The relationship between HER2 amplification in CRC cases with varying clinicopathological factors and prognosis and high-frequency mutated genes, TME immune cells, TLSs, and PD-L1 protein was analysed (Fig. 1).

Figure 1 Flow chart of the experimental procedure.

PSM, propensity score matching; WES, whole exome sequencing; TIME, tumor immune microenvironment; TLSs, tertiary lymphoid structures; IHC, immunohistochemistry; FISH, fluorescence in situ hybridization.

FISH verification of HER2 status in colorectal cancer

FISH detection was performed on 59 IHC-confirmed HER2-positive (IHC 1+, IHC 2+, IHC 3+) cases (Rüschoff et al., 2012). The Human HER2 Amplification Detection Kit (Wuhan Kanglu Biotechnology, Wuhan, China) was used to detect the HER2 gene with Rhodamine (RHO) fluorescein-labeled orange-red probe (HER2 probe) and fluorescein isothiocyanate (FITC)-labeled green probe (CEP17 probe) to detect the centromere sequence of chromosome 17. The criterion used to determine positive HER2 amplification was a HER2/CEP17 ratio ≥ 2 as in the HERACLES study (Sartore-Bianchi et al., 2016).

Gene mutation profiling in HACC using whole exome sequencing (WES)

The Magnetic Universal Genomic DNA extraction kit (Tiangen Biochemical Technology, Beijing, China) was used on ten 5 micron-thick CRC paraffin tissue section pieces to extract genomic DNA, amplify sample markers, enrich DNA and target genes by polymerase chain reaction (PCR). PCR will customize the capture area, using Ajitaikang’s liquid phase capture technology for enrichment. After enrichment, sequencing was performed on the Novaseq 6000 next-generation sequencing platform with a sequencing depth of 100 X.

Characteristic somatic mutations and their related signalling pathways in HACC

For screening, we investigated the following genes: (a) targeted therapy, prognosis, and resistance-related genes; (b) DNA-damage repair related genes; (c) human leukocyte antigen genes; (d) chemotherapy-related genes; (e) genetic susceptibility-related genes; details can be found in Data S1. The cBioPortal for Cancer Genomics database was used to query the pathogenic high-frequency mutations in HACC to obtain relevant signalling pathways.

Multiple fluorescence IHC (mIHC) for tumor-infiltrating leukocytes and tertiary lymphoid structures in HACC

Tumor-infiltrating leukocytes (including T cells, B cells, tumor associated macrophages, and NK cells) and tertiary lymphoid structures were detected using mIHC staining in the two groups of samples. Staining and imaging were performed using an automated staining instrument (BOND-RX Multiplex IHC Stainer, Leica, Wetzlar, Germany) and a quantitative pathology analysis system (Vectra Polaris, Akoya Biosciences, Marlborough, MA, USA). Antibody kits used included CD163: ab182422; CD56: ab75813; CD68: ab213363; CD8: ab178089; PD-L1: E1L3N (13684S); CK: ab7753; and S100: ab52642. All kits were from Bainuo Panorama Biotechnology (Beijing, China). HALO software (Indica Labs, Albuquerque, NM, USA) was used for tumor tissue and cell recognition, and for providing the density and percentages of cells positive for the different marker molecules in the intratumoural region (IT) and tumour rim (TR). Additionally, HALO was used to analyse the relationship between abnormal HER2 amplification and immune cell infiltration and diseases prognosis when compared with the control group.

IHC for PD-L1 status in HACC

IHC was used to detect the expression of PD-L1 protein in samples using Dako 22C3 kit (Agilent Technologies, Santa Clara, CA, USA) according to manufacturer’s instructions. PD-L1 protein expression results were independently observed and evaluated by two senior pathologists. PD-L1 protein is located in the cytoplasm or plasma and appears as yellow or brown particles. PD-L1 expression was determined by combining the tumor proportion score (TPS) and combined positive score (CPS). TPS: the number of positive tumor cells/total tumor cells of any intensity PD-L1 membrane staining ×100% CPS: (PD-L1 membrane staining positive tumor cells + PD-L1 positive tumor cells-associated immune cells)/total number of tumor cells ×100. CPS >1% was set as the threshold to indicate higher PD-L1 expression.

Statistical analysis

The results were analysed using SPSS 22.0 Statistics software (IBM, Armonk, NY, USA). Graphs were plotted using GraphPad Prism 8 software. Chi-square analysis, Fisher precision test and Pearson/Spearman correlation was used to analyse the correlation between age, sex, tumor site, pT stage, pN stage, pM, TNM stage, MMR, CEA, high-frequency mutant genes, and HER2 amplification in CRC. The association of HER2 amplification with TME immune cells, TLSs, and PD-L1 in negative CRC was analysed using Mann–Whitney U test. Survival was calculated using Kaplan–Meier and Cox regression analysis. Results were statistically significant when P < 0.05.

Results

The consistency of IHC and FISH in assessing HER-2 status for CRC

For IHC testing, of the 812 CRC cases collected, 753 cases were HER2-negative, and 59 were confirmed to be HER2-positive (IHC 1+, IHC 2+, IHC 3+). There were 18 cases of HER2 1+ with a detection rate of 2.22% (18/812), 18 cases of HER2 2+ with a detection rate of 2.22% (18/812), and 23 cases of HER2 3+ with a detection rate of 2.83% (23/812). The detection rate of IHC-confirmed HER2-positive was 7.27% (59/812).

For further FISH testing, all 59 IHC-confirmed HER2 positive (IHC 1+, IHC 2+ and IHC 3+) samples were subjected to FISH detection, and the total detection rate of FISH-validated HER2 amplification was 3.20% (26/812). Among them, 18 cases of IHC-HER2 1+ had a FISH-validated HER2 amplification ratio of 0 (0/18), 18 cases of IHC-HER2 2+ had a FISH-validated HER2 amplification ratio of 16.67% (3/18), and 23 cases of IHC-HER2 3+ had a FISH-validated HER2 amplification ratio of 100% (23/23). The consistent rate between IHC and FISH in assessing HER-2 status for CRC in our cohort was 44.07% (Fig. 2).

Figure 2 The consistency of IHC and FISH in assessing HER-2 status for CRC.

(A) Representative FISH results of HER2 amplification in CRC (magnification, x 200). Far left image shows tufted HER2 amplification. Near left image shows dotted HER2 amplification. Near right image shows positive staining control. Far right image represents negative staining control; (B) representative IHC results of HER2-positive CRC (magnification, x 200). Far left image shows IHC positive (3+) results of HER2 in CRC. Near left image shows IHC positive (2+) results of HER2 in CRC. Near right image shows positive staining control. Far right image represents negative staining control; (C) Flow chart of verification on HER2 amplification and the consistency of IHC and FISH in assessing HER-2 status for CRC.

HER2 status tested by IHC and FISH and its prognostic role in HER2-positive colorectal cancer

In order to investigate the correlation between HER2 amplification and primary tumor site, primary tumor site as located proximal (right) or distal (left) to the splenic flexure were categorized. HER2 expression rate in left side was higher than that of the right side, 80.77% (21/26) VS 19.23% (5/26), respectively. Compared with HER2-negative CRC, pM was positively correlated with HACC (correlation coefficient 0.092, P = 0.009). However, there was no difference in clinicopathological characteristics (age, sex, MMR status, CEA level, TNM stage, pT, pN, tumor site). Moreover, FISH-validated HER2 amplification status was not associated with CRC prognosis. Further studies found that IHC-HER2 1+, IHC-HER2 2+, IHC-HER2 (3+), and IHC-HER2 positive (1+, 2+ and 3+) were not related to the prognosis of CRC (P > 0.05), Table 1 and Fig. 3.

Figure 3 Kaplan–Meier survival analysis of HER2- positive and HER2-negative colorectal cancer tested by IHC and FISH.

(A1-A2) PFS and OS for FISH-HER2 amplified (n = 26) and HER2 negative (n = 786) colorectal cancer; (B1-B2) PFS and OS for IHC-HER2 positive (1+, 2+ and 3+, n = 59) and HER2 negative (n = 753) colorectal cancer; (C1-C2) PFS and OS for IHC-HER2 positive (2+, n = 18) and HER2 negative (n = 753) colorectal cancer; (D1-D2) PFS and OS for IHC-HER2 positive (3+, n = 23) and HER2 negative (n = 753) colorectal cancer. PFS, progression free survival; OS, overall survival.

Somatic mutations and their corresponding prognostic role in HACC

High frequency somatic gene mutations in HACC patients included TP53 (14/26), PIK3CA (8/26), EGFR (7/26), APC (7/26), ERBB3 (4/26), KRAS (4/26), ROS1 (4/26), FBXW7 (3/26), MAP2K1 (3/26), and AKT3 (3/26). There was no difference in gene mutation expression between HER2-amplified and HER2-negative CRC (P >  0.05). Most patients had mutations associated with RTK-RAS signalling pathway (EGFR, ERBB3, KRAS, MAP2K1, and ROS1), PIK signalling pathway (PIK3CA and AKT3), cell cycle signalling pathway (Cell Cycle, TP53, FBXW7), WNT signalling pathway (APC), NOTCH signalling pathway (FBXW7), and TP53 signalling pathway mutation (TP53).

The prognostic role of high-frequency of wild type and mutant gene mutation in HACC was analysed. Interestingly, overall survival (OS) of HACC with abnormal TP53 was worse than that of HER2-negative CRC (P = 0.038). The other remaining mutant genes were not related to the prognosis of HACC (P > 0.05), Tables 2 and 3 and Fig. 4.

Tumor-infiltrating leukocytes and their corresponding prognostic role in HACC

CD3 T (P = 0.039), FoxP3 T (P = 0.038), and CD3+CD4+ T (P = 0) cells were more abundant in the tumor rim of HACC than in HER2-negative CRC samples. Additionally, CD3+CD4+ T cells (P = 0) were more abundant in the intratumoral region of HACC than in HER2-negative CRC (Table 4 and Fig. 5). However, none of the tumor-infiltrating leukocytes was associated with the prognosis of HACC (Fig. 6).

Tertiary lymphoid structures and its prognostic role in HACC

Tertiary lymphoid structures in HACC (average of 0.07, values: 0–0.19) were higher than those in HER2-negative CRC (not applicable) (P = 0). Tertiary lymphoid structures did not correlate with the prognosis of HACC (Fig. 7.)

PD-L1 status and its prognostic role in HACC

According to TPS and CPS scores, the expression of PD-L1 protein in HACC was higher than that in HER2-negative cases (P = 0.023); however, its expression was not related to prognosis of HACC (Fig. 8).

Table 2 COX regression analysis of high-frequency gene mutations in HACC.

Gene	Mutant-type	Wild-type	
	PFS	OS	PFS	OS	
	HR (95%CI)	P	HR (95%CI)	P	HR (95%CI)	P	HR (95%CI)	P	
TP53	0.9325 (0.1866∼4.659)	0.932	0.4935 (0.09898∼2.460)	0.3889	1.462 (0.2780∼7.692)	0.6537	5.471(1.099∼27.24)	0.038	
PIK3CA	1.323 (0.1279∼13.69)	0.814	0.6492 (0.08287∼5.086)	0.6808	1.334 (0.3368∼5.282)	0.6816	2.676 (0.7093∼10.10)	0.1462	
EGFR	1.385 (0.1892∼10.14)	0.749	0.5751 (0.07211∼4.586)	0.6015	1.243 (0.3009∼5.135)	0.7636	3.018 (0.7795∼11.68)	0.1097	
APC	0.2292 (0.01889∼2.781)	0.247	2.054 (0.2579∼16.37)	0.4965	2.267 (0.6011∼8.551)	0.2268	2.146 (0.5490∼8.392)	0.2722	
KRAS	0.2303 (0.008907∼5.953)	0.376	0.2303 (0.008907∼5.)	0.3762	1.243 (0.3009∼5.135)	0.7636	3.018 (0.7795∼11.68)	0.1097	
ROS1	/	>0.999	/	>0.9999	1.696 (0.5132∼5.602)	0.3865	2.23 (0.6904∼7.202)	0.18	
ERBB3	/		/	/	1.448 (0.4141∼5.062)	0.5622	1.545 (0.4310∼5.540)	0.5041	
FBXW7	/	>0.999	0.7698 (0.07177∼8.256)	0.8289	1.353 (0.4202∼4.354)	0.6126	2.086 (0.5759∼7.558)	0.2628	
MAP2K1	/	>0.999	/	>0.9999	1.625 (0.4934∼5.353)	0.4246	2.19 (0.6809∼7.041)	0.1885	
AKT3	/	>0.999	3.794 (0.04104∼350.6)	0.5637	1.595 (0.4869∼5.222)	0.4407	1.987 (0.5874∼6.719)	0.2695	

Table 3 Gene mutations in HACC and HER2-negative colorectal cancer.

Genes	HER2 amplification (n = 26)	HER2 negative (n = 26)	X2	P	
TP53					
Mutant-type	14(53.85%)	10 (38.46%)	1.238	0.266	
Wild-type	12(46.15%)	16 (61.54%)	
PIK3CA					
Mutant-type	8(30.77%)	4 (15.38%)	0.975	0.332	
Wild-type	18(69.23%)	22 (84.62%)	
EGFR					
Mutant-type	7(26.92%)	9 (34.62%)	0.361	0.548	
Wild-type	19(73.08%)	17 (65.38%)	
APC					
Mutant-type	7(26.92%)	7(26.92%)	0	1	
Wild-type	19(73.08%)	19(73.08%)	
ERBB3					
Mutant-type	4 (15.38%)	0 (0)	/	0.110	
Wild-type	22 (84.62%)	26 (100%)	
ROS1					
Mutant-type	4 (15.38%)	1 (3.85%)	/	0.350	
Wild-type	22 (84.62%)	25 (96.15%)	
KRAS					
Mutant-type	4 (15.38%)	11 (42.31%)	0.885	0.347	
Wild-type	22 (84.62%)	15 (57.69%)	
FBXW7					
Mutant-type	3 (11.54%)	4 (15.38%)	/	1	
Wild-type	23 (88.46%)	22 (84.62%)	
MAP2K1					
Mutant-type	3 (11.54%)	1 (3.85%)	/	0.610	
Wild-type	23 (88.46%)	25 (96.15%)	
AKT3					
Mutant-type	3 (11.54%)	1 (3.85%)	/	0.610	
Wild-type	23 (88.46%)	25 (96.15%)	

Figure 4 Gene mutation of HACC and HER2-negative colorectal cancer, and Kaplan–Meier survival analysis of TP 53.

(A) Comparation of mutation frequency of 10 genes with the highest mutation rates in HACC (n = 26), HER2 negative colorectal cancer (n = 26) and The Cancer Genome Atlas (TCGA, PanCancer Atlas, gene mutation profiles from 534 patients with colorectal adenocarcinoma); (B) PFS and OS of HACC and HER2-negative colorectal cancer with wild-type TP53; (C) PFS and OS of HACC and HER2-negative colorectal cancer with mutant TP53.

Table 4 Tumor-infiltrating leukocytes and tertiary lymphoid structures in HACC and HER2 negative colorectal cancer.

Tumor-infiltrating leukocytes	HER2 amplification	HER2 negative	Z	P	
CD3 (IT)	144 (24∼461.25)	20 (0∼553.5)	−0.816	0.414	
CD3 (TR)	191 (86.5∼361.5)	27.5 (0∼541)	−2.066	0.039	
CD3 + CD4 + (IT)	26 (0∼100.5)	0 (0∼0)	−4.576	0	
CD3 + CD4 + (TR)	17.5 (4.75∼107.5)	0 (0∼0)	−5.863	0	
CD8 (IT)	15.5 (3.75∼39)	5.5 (0∼44.5)	−0.729	0.466	
CD8 (TR)	35.5 (14.5∼64.75)	4 (0∼70)	−1.952	0.051	
FoxP3 (IT)	33 (3∼68.5)	6.5 (0∼75.25)	−1.182	0.237	
FoxP3 (TR)	37.5 (5.75∼95.75)	1.5 (0∼59.75)	−2.071	0.038	
PD-1 + CD8 + (IT)	0 (0∼2.25)	0 (0∼1.25)	−0.439	0.66	
PD-1 + CD8 + (TR)	0 (0∼1)	0 (0∼1.5)	−0.478	0.632	
CD3 + CD4 + FoxP3 + (IT)	5 (0∼21.25)	0 (0∼14.5)	−1.61	0.107	
CD3 + CD4 + FoxP3 + (TR)	2 (0∼9.75)	0 (0∼9.75)	−1.194	0.233	
CD68 + CD163 + (IT)	2 (0∼7.5)	0 (0∼37.75)	−0.193	0.847	
CD68 + CD163 + (TR)	8.5 (2∼25.25)	0.5 (0∼70.5)	−1.445	0.148	
CD68 + CD163 - (IT)	28 (4.5∼60.75)	25 (0∼192.75)	−0.019	0.985	
CD68 + CD163 - (TR)	92.5 (30∼162.75)	48 (0∼186.75)	−1.181	0.238	
PD-L1 + CD68 + (IT)	0 (0∼3.25)	0.5 (0∼12.75)	−1.017	0.309	
PD-L1 + CD68 + (TR)	1 (0∼5.25)	0.5 (0∼11.5)	−0.524	0.6	
CD56bright (IT)	0 (0∼1)	0 (0∼1)	−0.263	0.792	
CD56bright (TR)	1.5 (0∼16)	0 (0∼5.25)	−1.303	0.193	
CD56dim (IT)	0 (0∼10.75)	0.5 (0∼21)	−0.998	0.318	
CD56dim (TR)	3.5 (0∼26.75)	2(0∼62)	−0.426	0.67	
CD20 (IT)	2 (0∼7)	0 (0∼6.5)	−1.048	0.295	
CD20 (TR)	9.5 (1∼22.5)	0 (0∼16)	−1.962	0.05	
TLSs	0.07 (0∼0.19)	N/A	−5.24	0	
Notes.

TLSs tertiary lymphoid structures

IT intratumoral region

TR tumor rim

Figure 5 Representative tumor-infiltrating leukocytes in HACC and HER2-negative colorectal cancer.

(A) Merged image of tumor-infiltrating leukocytes staining in tumor tissues from HER2-amplified colorectal cancer, CD8 (pink), CD68 (cyan), CD163 (red); (B) merged image of tumor-infiltrating leukocytes staining in tumor tissues from HER2-amplified colorectal cancer, CD3 (pink), CD4 (red), CD20 (green), CD56 (cyan), FoxP3 (yellow); (C) positive staining controls of tumor-infiltrating leukocytes, CD8 (pink), CD68 (cyan), CD163 (red), CD3 (pink), CD4 (red), CD20 (green), CD56 (cyan), FoxP3 (yellow); (D) negative staining control of tumor-infiltrating leukocytes; (E) tumor-infiltrating leukocytes cell counts of intratumoral region (IT) in HACC and HER2-negative colorectal cancer; (F) tumor-infiltrating leukocytes cell counts of tumor rim (TR) in HACC and HER2-negative colorectal cancer.

Figure 6 Kaplan–Meier curves for tumor-infiltrating leukocytes in IT (intratumoral region) and TR (tumor rim).

(A1-A2) PFS and OS for CD3 T cells (TR) in HACC; (B1-B2) PFS and OS for FoxP3 T cells (TR) in HACC; (C1-C2) PFS and OS for CD3 +CD4+ T cells (IT) in HACC; (D1-D2) PFS and OS for CD3+CD4+ T cells (TR) in HACC.

Figure 7 Cell counts and Kaplan–Meier curves of tertiary lymphoid structures in HACC and HER2-negative colorectal cancer.

(A) Cell counts and the corresponding ratio of tertiary lymphoid structures in HACC and HER2-negative colorectal cancer; (B-C) PFS and OS for tertiary lymphoid structures in HACC.

Figure 8 PD-L1 expression by Immunohistochemistry staining and Kaplan–Meier survival analysis of PD-L1 expression based on CPS.

(A) IHC staining of PD-L1 (brown) on tumor cells in HACC. Left image shows IHC staining of PD-L1 in HACC, middle image is the positive control, and right image is the negative control; (B) Kaplan–Meier survival analysis of PD-L1 expression based on CPS (threshold: >1%). Left: progression-free survival (PFS) for high and low PD-L1 expression in HACC. Right: overall survival (OS) for high and low PD-L1 expression in HACC.

Discussion

Despite a rate of 2–11% (Guarini et al., 2021; Ivanova et al., 2022; Siena et al., 2018) of HER2 positive expression in CRC, the association between HER2-positive status and the prognosis of CRC is still controversial. Additionally, the characteristics of high-frequency gene mutations and TME immune cells in HACC, and whether it can benefit from ICIs treatment are unknown.

In this study, the preliminary screening of HER2 status of 812 postoperative samples of CRC by IHC yielded a positive detection rate of 7.27% (59/812). Then, these IHC-HER2 positive samples were tested by FISH, and the detection rate of FISH-HER2 was 3.2% (26/812). The consistency rate between IHC and FISH was 44.07% (26/59). According to the above results, and in line with reports such as the HERACLES study, in CRC, IHC detection provides preliminary screening for HER2 status, while FISH is the gold standard for detecting HER2. It is worth noting that we found IHC-HER2 (3+) and FISH to have a 100% detection agreement rate, while IHC-HER2 (2+) and IHC-HER2 (1+) had agreement rates of 16.67% and 0%, respectively. This suggests that IHC-HER2 (3+) can be determined directly as positive HER2 amplification, while IHC-HER2 (2+) and IHC-HER2 (1+) need FISH confirmation to determine HER2 status. This is consistent with the National Comprehensive Cancer Network (NCCN) and The Chinese Society of Clinical Oncology (CSCO) (Benson et al., 2021; Weng et al., 2020). Moreover, the application of next-generation sequencing (NGS) confirmed HER2 amplification detection. However, considering the high cost of NGS and the results from other methods in this study, we believe that FISH should remain as the benchmark method for HER2 positive status detection (Cenaj et al., 2019; Fujii et al., 2020).

Due to the controversy between HER2-positive status and CRC prognosis, we analysed the association between IHC-HER2-positive and FISH-HER2 amplification with CRC prognosis. In this study, the status of 1+, 2+, or 3+ detected in IHC did not correlate with the prognosis of CRC, which is consistent with previous reports (Kilicarslan et al., 2018). Additionally, HER2 amplification status verified by FISH was not an indicator of CRC prognosis.

Simultaneously, in the analysis of the association between HACC and clinicopathological features, only pM was positively correlated with HACC. That is, HER2 amplification was associated with distant metastasis of CRC.

To understand whether HACC has characteristic high-frequency gene mutations, we performed WES for HACC. The results showed that TP53, PIK3CA, EGFR, APC, ERBB3, KRAS, ROS1, FBXW7, MAP2K1, AKT3 appeared at high frequency in HACC, and the detection rate of these genes was consistent with that of the control group and the gene mutation profiles from 534 patients with COAD were obtained from The Cancer Genome Atlas (TCGA, PanCancer Atlas) database (Cancer Genome Atlas, 2012). Through cBioPortal database analysis, we found that these genes mainly play a role in RTK-RAS, PIK, CELL CYCLE, WNT, NOTCH, and TP53 signalling pathways.

Furthermore, we analysed the correlation between CRC prognosis, HER2 amplification and the mutant and wild-type states of the genes mentioned above. We found that the OS of HACC with a wild-type TP53 state was worse than that in the control group. There was no relationship between CRC prognosis, amplification of HER2, and the status of the remaining genes. TP53 is a factor in cancer development and tumor progression and plays an important role in sensitivity in chemotherapy and radiation therapy (Chang et al., 2013; El-Deiry et al., 2015; Koncina et al., 2020; Nakayama & Oshima, 2019; Tomicic, Dawood & Efferth, 2021). In this study, we found for the first time that HER2 amplification and a wild-type TP53 state is associated with poor CRC prognosis, suggesting that TP53 gene mutation may be a potential protective mechanism for HACC.

Next, we analysed TME immune cells in HACC by mIHC, and found that CD3 T cells, FoxP3 T cells, CD3+CD4+ T cells, and TLSs were higher in HACC than in the control group. Furthermore, we analysed the prognostic association between the above cells, TLSs and HER2 CRC, and found that none of them were indicators of immune prognosis. While the degree of T cell invasion and TLSs were not directly related to the prognosis of HACC, it should be noted that they all showed different degrees of invasion in these cases. This suggests that ICIs can be considered for treatment in cases of HER2 amplification enriched with effector T cells or TLSs. Since this study is retrospective, the patients were not treated with ICIs. However, in subsequent treatment processes, it is possible that ICIs for effector T cells and TLSs will be considered for these patients. For example, CD3 T cells are the main recognition unit in recognizing antigens; previous studies had shown that CD3 x anti-HER2 bispecific antibodies increased the cytotoxic activity of HER2-positive tumor cells and had significant antitumour effects (Han et al., 2014; Ren et al., 2012). CD3+CD4+ T cells, which recognize antigens (Osman et al., 2021; Xiaomei, Chunyan & Zonghong, 2020); Foxp3 T cells has an immunosuppressive effect and promotes disease progression (Hori, Nomura & Sakaguchi, 2003; Kuwahara et al., 2019; Zhe, 2020).

Lastly, several studies have shown that PD-L1 expression is upregulated in a variety of malignancies, inducing tumor immune evasion (He et al., 2021; Yang et al., 2022). In this study, PD-L1 expression was higher in HACC than in HER2-negative CRC. However, its expression did not correlate with HER2 amplification prognosis. This shows that PD-L1 protein may not be a potential target for the treatment of HACC.

In summary, a total of 26 HER2-amplified samples were collected in this study, and the agreement rate between IHC initial screening and FISH verification was 44.07%. This confirms FISH to be the gold standard for determining HER2 amplification in CRC. In this study, HER2 status was not an indicator for CRC prognosis. Further, there are few reports on the characteristics of gene mutations and TME immune cells in HACC. In this study, we explored these aspects and found that HER2 amplification and a wild-type TP53 state had poor CRC prognosis. Additionally, TME immune cells, tertiary lymphoid structures and PD-L1 protein were found to be higher in HACC than those in negative control cases, but they were not related to prognosis.

Limitations of this study include possible statistical bias owing to the small sample size and the relatively short follow-up time. For the former, the data was strictly selected by the propensity score matching method to avoid selection bias as much as possible. Additionally, for patients with HACC, trastuzumab treatment was recommended in previous studies. However, since this is a retrospective study, we were not able to introduce this treatment, and thus we could not evaluate trastuzumab effectiveness in HACC.

Supplemental Information

Supplemental Information 1 Raw data exported from Tables 1–4 and Figs. 2–8

Click here for additional data file.

Supplemental Information 2 CRC associated genes

Click here for additional data file.

Supplemental Information 3 HER2 amplification colorectal cancer associated genes exported from Table 3

Click here for additional data file.

This study is a joint effort of many investigators and staff members, and their contribution is gratefully acknowledged. We especially thank all patients who participated in this study.

Additional Information and Declarations

Competing Interests

Author Contributions

Human Ethics

Field Study Permissions

Data Availability

The authors declare there are no competing interests.

Xiao-Ting Liu conceived and designed the experiments, performed the experiments, analyzed the data, prepared figures and/or tables, authored or reviewed drafts of the article, and approved the final draft.

Zhi-Yong Kou performed the experiments, prepared figures and/or tables, authored or reviewed drafts of the article, and approved the final draft.

Hushan Zhang performed the experiments, prepared figures and/or tables, authored or reviewed drafts of the article, and approved the final draft.

Jian Dong performed the experiments, analyzed the data, prepared figures and/or tables, authored or reviewed drafts of the article, and approved the final draft.

Jian-Hua Zhang analyzed the data, prepared figures and/or tables, and approved the final draft.

You-Jun Peng analyzed the data, prepared figures and/or tables, and approved the final draft.

Shu min Ma analyzed the data, prepared figures and/or tables, and approved the final draft.

Lei Liang analyzed the data, prepared figures and/or tables, and approved the final draft.

Xuan-Yu Meng analyzed the data, prepared figures and/or tables, and approved the final draft.

Yuan Zhou analyzed the data, prepared figures and/or tables, and approved the final draft.

Jun Yang conceived and designed the experiments, performed the experiments, analyzed the data, prepared figures and/or tables, authored or reviewed drafts of the article, and approved the final draft.

The following information was supplied relating to ethical approvals:

Research Ethics Board of The First Affiliated Hospital of Kunming Medical University (Ethical Application Ref: 2022-L271).

Field experiments were approved by Research Ethics Board of Kunming Medical University.

The following information was supplied regarding data availability:

The raw measurements are available in the Supplementary File.

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
