# Peer review of "Somatic mutation profiling, tumor-infiltrating leukocytes, tertiary lymphoid structures and PD-L1 protein expression in HER2-amplified colorectal cancer"

_PeerJ, doi:10.7717/peerj.15261_

## Round 0.1 · original submission · Minor Revisions

I have received comments from three independent experts in the field, who have all raised some issues that you will need to respond to by modifying the manuscript accordingly. Some of these are scientific points and some relate to the language you have used. Please take this opportunity to improve the manuscript content.

Reviewer 1 ·

Basic reporting

This article manuscript aims to explore the relationship between HER2 amplification in colorectal cancer cases with varying clinicopathological factors and prognosis and gene mutations, tumor-infiltrating leukocytes, TLSs, and PD-L1 protein expression. The language used in this manuscript was clear, unambiguous, and professional.
The introduction and background offered the readers a good amount of information. However, for some points, if supplemented with more information, it would be easier for the readers to understand. For example, the authors mentioned IHC-confirmed HER2-positive (IHC 1+, IHC 2+, IHC 3+) in line 115. But the authors did not give enough information about the differences between them.
The figures and tables are relevant, high quality, and well-labeled. However, when the authors described the patient characteristics (line 83-102), the authors can cite Table 1. If so, it would be easier for the readers. Similar improvements can be also done by citing Figure 1 around line 176-182.
In line 185, the authors mentioned left side and right side, but the authors did not clarify the definition of left and right in the context.
In line 176-182 and 232-236, the author compared IHC and FISH methods in identifying HER2-positive colorectal cancer. Although the authors pointed out FISH method is more accurate, the explanation of the mechanism for the reason for the differences is not sufficient.
Although the abbreviation of colorectal cancer (CRC) was mentioned in line 21, the authors still mention the full names and abbreviations interchangeably. It is a little confusing and can be improved. Similar problem with HACC abbreviations.
Some small formatting problems exist in the manuscript.
Line 156, missing space between % and CPS.
Line 286 missing space in front of “In this study”

Experimental design

The experiments are well-designed. The research questions are well-defined, relevant and meaningful. The methods were described in sufficient detail.

Validity of the findings

The authors tried to find correlations between HER2 amplification in colorectal cancer cases with varying clinicopathological factors and prognosis and gene mutations, tumor-infiltrating leukocytes, TLSs, and PD-L1 protein expression. The authors found pM was positively correlated with HER-2 amplified colorectal cancer. However, possibly due to the small cohort, there was no difference in clinicopathological characteristics between HER2-negative and amplified colorectal cancer. There was no difference in gene mutation expression between HER2-amplified and negative colorectal cancer. Overall survival of HER2-amplified colorectal cancer with abnormal TP53 was worse than that of HER2-negative colorectal cancer. None of the tumor-infiltrating leukocytes was associated with the prognosis of HER2-amplified colorectal cancer. And tertiary lymphoid structures did not correlate with the prognosis of HER2-amplified colorectal cancer. Meanwhile, PD-L1 expression was not related to the prognosis of HER2-amplified colorectal cancer.
Although not many conclusions are reached and limited mechanisms are investigated, the article offered good insights into HER2-amplified colorectal cancer. It can further facilitate clinical research on HER2-amplified colorectal cancer.

Reviewer 2 ·

Basic reporting

The focus of this manuscript is to explore the characteristics of the somatic mutation
spectrum, tumor-infiltrating leukocytes, tertiary lymphoid structures, and PD-L1 protein expression in HER2-ampliûed colorectal cancer. This study greatly advances the knowledge in the field. The manuscript is overall well-written. I only have a few suggestions for improvement.

1. The Introduction part is inadequate. The authors only briefly introduced the motivation they decided to explore the somatic mutation spectrum, tumor-infiltrating leukocytes, and tertiary lymphoid structures in HER2-ampliûed colorectal cancer. The reason for investigating the expression of PD-L1 protein in HER2-ampliûed colorectal cancer is not clear. The authors should also introduce HER2, the effect of HER2-positive status, and previous studies on HER2-positive colorectal cancer patients or patients with other cancer types.

2. The English language should be improved to ensure that an international audience can clearly understand your text. Some examples where the language could be improved include: Lines 32-33: For somatic mutation profiling, there are no differences between HACC and HER2-negative colorectal cancer, however TP 53, which had a strong correlation to poor prognosis in HACC.
Lines 34-35: For the immune cells and PD-L1 expression, T cells and TLSs in Tumor immune microenvironment, as well as PD-L1 expression were higher in HACC than HER2-negative control.

Experimental design

From line 176 to 182, the authors used FISH assay to test the 59 IHC-confirmed HER2 positive samples. They reported that the total detection rate of FISH-validated HER2 amplification was 3.20% (26/812). However, since FISH assay was only used on the 59 IHC-confirmed HER2 positive samples and the other 753 samples were not detected by FISH, could the detection rate of FISH-validated HER2 amplification be calculated as 26/812?

Validity of the findings

The manuscript conducted many assays and obtained a lot of results. Although the results were presented in the tables and figures, the authors did not give sufficient interpretation of the results, the meanings and indications of the results should be explained in more details.

Reviewer 3 ·

Basic reporting

line56-60: Verbose, please make the sentence clear
line210: 'whether in the intratumoural region or tumour rim', expression is not clear.

Experimental design

1. The authors only compared the gene mutation of HACC patient with non-HACC patient, there is no information about the transcriptome difference.

Validity of the findings

1. HACC patient with WT TP53 has lower OS compared to HACC patient with mutant TP53, is there any explanation why TP53 mutation benefit the HACC patient survival?
2. According to the result, Her2 amplification is associated with metastasis, does HACC patients have any genes expression or gene mutation related with the metastasis signaling?

---

## Round 0.2 · accepted · Accept

Many thanks for submitting your revised manuscript. I have received reviewer comments (attached), and they are supportive of your article being accepted for publication.

Reviewer 1 ·

Basic reporting

The authors have responded to the points in the comments appropriately, and the manuscript was edited accordingly.

Experimental design

No comment.

Validity of the findings

No comment.